# MEDA: Medical-Oriented Activation Editing for Hallucination Mitigation in Medical Large Vision-Language Model

Tianbo Wang [1 2]   Yuqing Ma [3 2]   Lingyan Meng [1]   Zhange Zhang [3 2]   Kewei Liao [1 2]   Jian Yang [1 2]   Simin Li [1 2 4]
Jinyang Guo [3 2]   Xianglong Liu [1 2]

## Abstract

Despite notable advances in automated medical image interpretation, Medical Large Vision-Language Models (Med-LVLMs) continue to suffer from severe hallucinations, posing critical safety risks in clinical deployment. Editing LVLM activations has shown promise for mitigating hallucination with minimal cost. However, due to the requirements of medical domain expertise, existing methods struggle to capture imaging manifestations and diagnostic principles that are critical for clinical interpretation, thereby limiting their effectiveness. To address these limitations, we propose the first MEDical-oriented Activation Editing (MEDA) method by integrating Query-decisive Manifestation Steering (QMS) and Principle-driven Diagnosis Induction (PDI) to promote Med-LVLM's expertise elicitation. Specifically, QMS retrieves positive query-decisive imaging manifestations as trusted guidance for activation steering, while PDI constructs positive principle-embedded diagnostic prompts to induce expert-like clinical reasoning. Extensive experiments across multiple modalities demonstrate that MEDA efficiently improves the response factuality for both VQA and report generation tasks, achieving up to 10.6% improvement on IU-Xray, while exhibiting generalization and few-shot robustness for practical application.

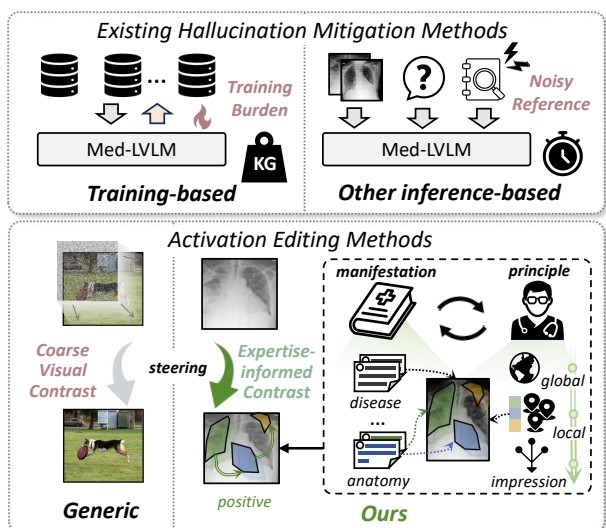

*Figure 1.* Comparisons of MEDA with other medical hallucination mitigation approaches, including training-based, other inference-based, and generic activation editing methods.

## 1. Introduction

With advances in automated and scalable multimodal modeling, Large Vision-Language Models (LVLMs) have demonstrated promising performance on medical tasks (Li et al., 2023; Chen et al., 2024b), such as medical visual question answering (Hu et al., 2024; Guo et al., 2025) and report generation (Lee et al., 2024; Wang et al., 2025a). However, LVLMs are particularly prone to hallucinations and generate erroneous responses in medical scenarios (Kim et al., 2025; Zhu et al., 2025b), where images exhibit weaker visual cues compared to natural images (Huang et al., 2021), and require domain expertise for interpretation (Xia et al., 2025). This issue can pose serious safety risks, hampering the reliable deployment in real-world clinical scenarios.

The severity of hallucinations in medical LVLMs (Med-LVLMs) has drawn broad attention, with existing mitigation approaches broadly categorized into training-based and inference-based methods. Training-based methods rely on fine-tuning (Xie et al., 2025; Zhu et al., 2025a) with curated medical datasets, while their clinical applications are

[1]School of Computer Science and Engineering, Beihang University [2]State Key Laboratory of Complex & Critical Software Environment [3]Institute of Artificial Intelligence, Beihang University [4]The Chinese University of Hong Kong. Correspondence to: Yuqing Ma <mayuqing@buaa.edu.cn>.

*Proceedings of the 43rd International Conference on Machine Learning*, Seoul, South Korea. PMLR 306, 2026. Copyright 2026 by the author(s).

constrained by large-scale data and overwhelming training burden. Inference-based methods avoid retraining by incorporating external information (*e.g.*, retrieval-augmented generation) (Xia et al., 2024; Lu et al., 2025) or alternative reasoning strategies (Savage et al., 2024; Liu et al., 2024b) at inference time, while introducing noisy references (Xia et al., 2025) and increased inference cost.

By encouraging LVLMs to better exploit their internal knowledge and capability (Li et al., 2024a), inference-time activation editing techniques (Chen et al., 2024a; Wang et al., 2025b) effectively mitigate hallucinations in generic LVLMs while exhibiting robustness to noise and incurring minimal inference costs (Chen et al., 2025; Wang et al., 2026). Specifically, these techniques directly optimize LVLMs' activations using visually contrastive steering vectors, which are derived from trusted and untrusted samples. For instance, VTI (Liu et al., 2025) derives steering vectors by contrasting stable and noise-perturbed visual features to reinforce consistent visual knowledge encoded in LVLM. VISTA (Li et al., 2025b) constructs vectors from trusted image-present and untrusted image-absent contexts, thereby promoting the utilization of visual–semantic knowledge.

However, as shown in Figure 1, existing generic editing methods struggle to capture medical expertise solely from contrasts in image appearance, thereby limiting their effectiveness for mitigating hallucination. For example, imaging manifestations encode disease-specific visual patterns (*e.g.*, pulmonary opacities in pneumonia) that are not reliably captured by image-level contrasts. It guides the model to emphasize semantically salient cues beyond superficial pixel differences, thereby improving visual perception of medical images. Additionally, diagnostic principles reflect clinicians' experience-informed diagnostic strategies (including reading priorities, *etc.*) in image interpretation, which is an empirical form of expert reasoning. Incorporating these principles facilitates the integration of clinical diagnosis, enhancing the cognitive interpretation of medical images.

In this paper, we propose **MED**ical-oriented **A**ctivation Editing (**MEDA**), the first activation editing paradigm designed specifically for medical imaging. MEDA integrates **Q**uery-decisive **M**anifestation **S**teering (**QMS**) with **P**rinciple-driven **D**iagnosis **I**nduction (**PDI**), thereby promoting elicitation of non-obvious medical expertise and enhancing both perceptual and cognitive interpretation of medical images. Specifically, QMS performs precise entity-grounded retrieval of positive query-decisive imaging manifestations, and contrasts them with the original activations to obtain manifestation-aware steering vectors. By editing internal activations to elicit correct manifestation knowledge, QMS reliably guides Med-LVLMs toward accurate visual perception of medical images. Simultaneously, PDI encodes textbook-recommended diagnostic principles into

templates to construct positive diagnostic prompts, and derive principle-aware editing vectors by contrasting their activations against original activations. Therefore, PDI can induce expert-like cognitive patterns in Med-LVLMs' generation behavior guided by principle-aware vectors. Finally, MEDA performs cooperative steering by jointly applying manifestation-aware and principle-aware vectors, thereby achieving effective hallucination mitigation.

Experimental results across six datasets spanning three imaging modalities and six LVLMs demonstrate that MEDA effectively improves the factuality of both medical VQA and report generation answers with minimal cost, achieving up to 10.6% improvement on IU-Xray. MEDA also exhibits generalizability across multi-source datasets and maintains effective performance with few samples, highlighting its practical applicability. In summary:

- We propose MEDA, the first efficient activation editing paradigm tailored to medical scenarios, which steers LVLMs toward medically grounded expertise for mitigating hallucination.

- We introduce QMS, which retrieves query-decisive imaging manifestations to guide activation editing for enhanced medical image perception.

- We propose PDI, which incorporates textbook-recommended diagnostic principles to steer Med-LVLMs with chain-structured clinical diagnosis.

- Extensive experiments across six datasets spanning three imaging modalities and six LVLMs demonstrate that MEDA achieves superior factuality improvements with minimal cost, while exhibiting strong cross-dataset generalization and effectiveness with few samples, highlighting its practical applicability.

## 2. Related Works

**Mitigating Hallucination of Med-LVLMs**  Existing approaches for mitigating hallucinations in Med-LVLMs can be broadly categorized into training-based methods and inference-based methods. Training-based methods employ fine-tuning strategies (Xie et al., 2025; Nguyen et al., 2025; Zhu et al., 2025a) on constructed medical datasets to improve visual interpretation and reasoning. However, they require substantial amounts of high-quality training data and incur high computational costs, which limit their applicability in data-scarce clinical scenarios. To avoid the cost of retraining, inference-based methods improve output factuality by incorporating external reference (*e.g.*, retrieval-augmented generation (RAG) (Xia et al., 2025; Lu et al., 2025)) or introducing alternative reasoning paradigms (*e.g.*, chain-of-thought prompting (Savage et al., 2024; Liu et al., 2024b)) during inference. Nevertheless, naively introducing

extensive reference often brings significant noise that may mislead Med-LVLMs (Xia et al., 2025), while numerous retrieved documents and lengthy reasoning chains further exacerbate inference computational overhead.

**Activation Editing Technique**  Inference-time activation editing techniques (Li et al., 2024a; Zhang et al., 2024; Chen et al., 2024a) have shown promise in addressing hallucinations in generic LVLMs (Chen et al., 2025; Liu et al., 2025). These techniques encourage LVLMs to better exploit their internal knowledge and capability (Li et al., 2024a) by editing activations with carefully designed steering vectors, enabling efficient hallucination mitigation without introducing explicit input noise. For instance, VTI (Liu et al., 2025) derives steering vectors by contrasting stable and noise-perturbed visual features, and intervenes in activations to reinforce consistent visual knowledge encoded in LVLM. VISTA (Li et al., 2025b) constructs vectors from image-present and image-absent contexts in activation space, thereby promoting the utilization of genuine visual–semantic knowledge during generation. However, these generic editing methods struggle to capture professional and empirical medical knowledge solely from contrasts in image appearance.

## 3. Methodology

To effectively reduce hallucination in Med-LVLMs, we introduce the first medical-oriented activation editing (MEDA) method. As shown in Figure 2, MEDA leverages Query-decisive Manifestation Steering (QMS) and Principle-driven Diagnosis Induction (PDI) to steer Med-LVLMs with medical expertise of imaging manifestation and diagnostic principle, thereby enhancing both perceptual and cognitive interpretation of medical images. In this section, we first present preliminary in Section 3.1, and elaborate on QMS in Section 3.2 and PDI in Section 3.3, respectively.

### 3.1. Preliminary

Given an LVLM $\mathcal{M}$ encoded with rich pretrained knowledge, the model can process a query composed of an image-question pair $\langle x, q \rangle$, and generate an answer $\hat{a} = \mathcal{M}(x, q)$. During the forward pass of $\mathcal{M}$, the image-question pair $\langle x, q \rangle$ is tokenized and subsequently passed through $L$ decoding layers with $H$-head self-attention, yielding the hidden states at each layer as $\mathbf{o}^l$:

$$\mathbf{o}^{l+1} = \mathbf{o}^l + \text{Concat}_{h=1}^{H}(\mathbf{z}^{(l,h)}) \cdot W_o^l, \qquad (1)$$

where $\mathbf{z}^{(l,h)} = \text{Attn}^{(l,h)}(\mathbf{o}^l)$ denotes the internal activation after self-attention operation of the $h$-th head at the $l$-th layer, $W_o^l$ is an output projection matrix. However, $\mathcal{M}$ may fail to appropriately elicit relevant knowledge, rendering the generated answer $\hat{a}$ to deviate from the ground-truth answer

$a$. Therefore, sparse interventions on the internal activations have been designed by activation editing to guide the model toward producing non-hallucinatory outputs.

Typically, for each head $(l, h)$, these methods compute steering vector $\mathbf{d}^{(l,h)}$ by averaging the differences between trusted visual activation $\mathbf{z}^{(l,h)+}$ and untrusted visual activation $\mathbf{z}^{(l,h)-}$ over an image set $\mathbf{X}$. The vector is then applied to the activation during inference as follows:

$$\hat{\mathbf{z}}^{(l,h)} = \mathbf{z}^{(l,h)} + \alpha \cdot \mathbf{d}^{(l,h)}, \quad \text{where}$$
$$\mathbf{d}^{(l,h)} = \frac{1}{|\mathbf{X}|} \sum_{\mathbf{X}} \left( \mathbf{z}^{(l,h)+} - \mathbf{z}^{(l,h)-} \right). \qquad (2)$$

In Eq.2, $\hat{\mathbf{z}}^{(l,h)}$ denotes the edited activations and $\alpha$ denotes the editing intensity. However, previous methods typically construct $\mathbf{d}$ [1] through coarse visual contrasts, failing to capture medical expertise that cannot be directly observed from image appearance. In contrast, our QMS in Section 3.2 and PDI in Section 3.3 explicitly incorporate two crucial forms of medical expertise into the positive samples, *i.e.* imaging manifestation and diagnostic principles, thereby inducing steering directions that elicit the Med-LVLM's clinically grounded knowledge utilization for factual perception and reasoning, respectively.

### 3.2. Query-decisive Manifestation Steering

We first introduce Query-decisive Manifestation Steering (QMS) to capture trusted manifestation guidance ignored by previous methods. To better address clinically grounded queries, QMS identifies the most diagnosis-relevant entity under a predefined extraction priority, and uses this entity as the keyword to retrieve the positive manifestation guidance from authoritative databases. Based on this guidance, QMS derives the manifestation-aware steering direction for activation editing, thereby eliciting Med-LVLMs' knowledge utilization to improve response factuality.

**Diagnosis-relevant Entity Extraction**  Motivated by the observation that entity–query relevance varies across entity categories, QMS instructs a powerful model $\mathcal{F}$ (*e.g.* GPT-4 (Achiam et al., 2023)) to explicitly enforce a category-aware extraction principle, thereby identifying the most diagnostically informative entity from given question-answer pair.

Specifically, we first define three targeted categories: $\mathbf{C} = \{c_d, c_a, c_o\}$, where $c_d$ denotes disease, $c_a$ denotes anatomy, and $c_o$ represents other entities beyond disease and anatomy (*e.g.*, *pacemaker*). Empirically, the diagnostic relevance of these entity categories decreases progressively (*e.g.* disease *"lung cancer"* is more relevant than anatomy *"lung"* for the

---
[1]Due to the identical operation, we omit the layer $l$ and head $h$ indices in the upper right corner for all activation and vector symbols in the following Sections to simplify the notation.

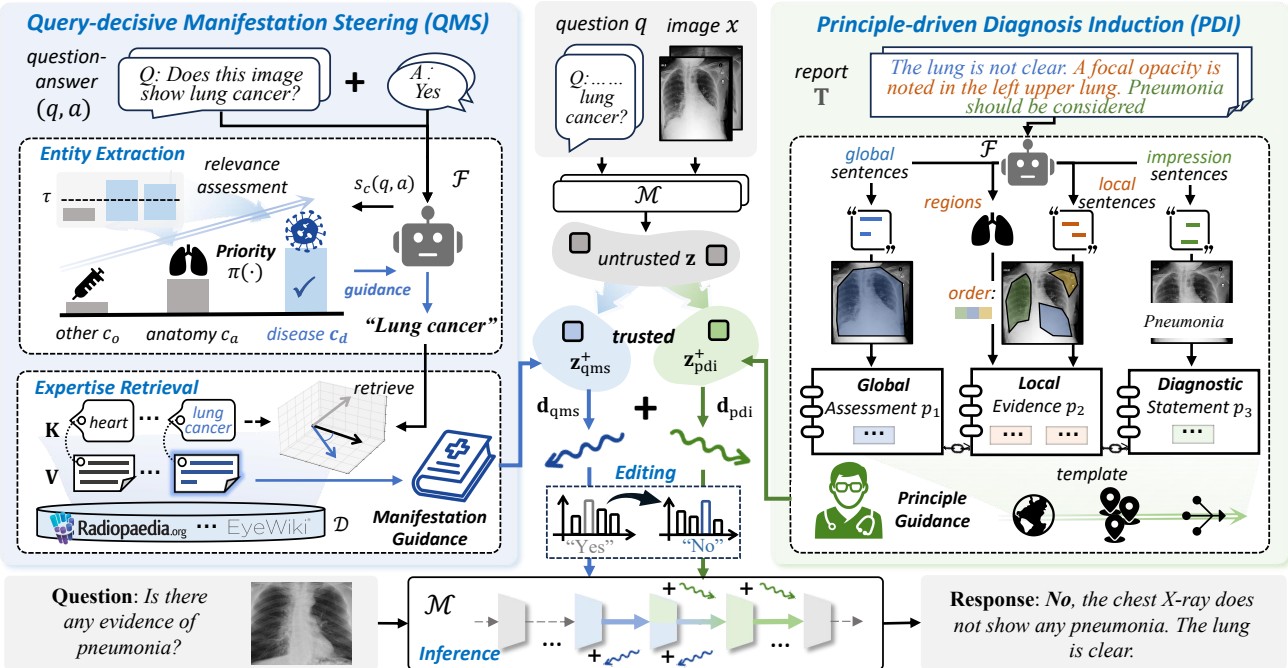

*Figure 2.* An Overview of proposed MEDA. MEDA leverages QMS and PDI to steer Med-LVLMs with medical expertise of imaging manifestation and diagnostic principles, thereby enhancing both perceptual and cognitive interpretation of medical images.

query *"Is there any evidence of lung cancer?"*), which can be formalized by a deterministic priority weight function $\pi(\cdot)$ such that $\pi(c_d) > \pi(c_a) > \pi(c_o)$. Accordingly, we aim to leverage $\pi(\cdot)$ to determine which entity category should be prioritized given the question–answer pair $(q, a)$, thereby enabling more precise diagnosis-relevant entity extraction. To achieve this, we first prompt $\mathcal{F}$ with category-scoring instruction $I_{sco}$ to score the relevance to each category $c \in \mathbf{C}$. Subsequently, $\mathcal{F}$ selects the determined category according to $\pi(\cdot)$ among all potential categories exceeding a relevance threshold $\tau$, and then applies an entity-generation instruction $I_{ent}$ to obtain category-specific entity. It is noted that if no category scores over $\tau$ (*e.g.*, *"What is the modality of the radiograph?"*), the entity extracted by $\mathcal{F}$ is set to None. We summarize the whole procedure using the formula:

$$e = \mathcal{F}\left(I_{ent}; [q, a], \underset{c \in \mathbf{C}}{\arg\max}\big(\mathbb{I}[s_c(q, a) > \tau] \cdot \pi(c)\big)\right) \quad (3)$$

where $s_c(q, a) = \mathcal{F}(I_{sco}; [q, a], c)$ denotes the relevance score of category $c$, $\mathbb{I}[\cdot]$ denotes the indicator function.

**Entity-grounded Expertise Retrieval**    Grounded in the entity $e$, we can retrieve query-decisive imaging manifestation $m$ from authoritative medical imaging databases $\mathcal{D}$ (*e.g.* Radiopaedia (Gaillard et al., 2011)), where $\mathcal{D} : \mathbf{K} \to \mathbf{V}$ is formalized as a dictionary-style mapping from medical terms $\mathbf{K}$ to their corresponding manifestation expertise $\mathbf{V}$. Specifically, QMS first aligns $e$ to medical terms $\mathbf{K}$ to ensure that each entity can be queried in databases $\mathcal{D}$. It adopts

the pretrained Bio-ClinicalBERT (Alsentzer et al., 2019) as an encoder $\mathcal{E}$ to embed both $e$ and all candidate terms in $\mathbf{K}$, and assign $e$ to the candidate with the highest cosine similarity. As in entity extraction, we also apply the threshold $\tau$ to control retrieval validity: if the maximum similarity falls below $\tau$ (*e.g.*, when $\mathbf{K}$ has no appropriate entry for the extracted entity $e$), we discard the pair $(q, a)$ from the construction data. Finally, we curate the primary introduction and manifestation descriptions from $\mathcal{D}$ as the retrieved expertise $m$. The whole procedure can be formulated as:

$$m = \begin{cases} \mathcal{D}\left(\underset{k \in \mathbf{K}}{\arg\max} \, \text{sim}(e, k)\right), & \underset{k \in \mathbf{K}}{\max} \, \text{sim}(e, k) \geq \tau \\ \varnothing, & \underset{k \in \mathbf{K}}{\max} \, \text{sim}(e, k) < \tau \end{cases},$$
$$(4)$$

where $\text{sim}(e, k) = \frac{\mathcal{E}(e)^{\top}\mathcal{E}(k)}{\|\mathcal{E}(e)\|\|\mathcal{E}(k)\|}$ denotes similarity score.

**Manifestation-aware Vector Steering**    Subsequently, QMS injects positive manifestation expertise $m$ into the original input as trusted guidance, and constructs trusted-untrusted sample pairs $\langle(x, m, q), (x, q)\rangle$, which are then fed into Med-LVLM $\mathcal{M}$ to obtain the trusted-untrusted activation pairs $\langle \mathbf{z}_{qms}^{+}, \mathbf{z} \rangle$. Therefore, we can directly model the manifestation-aware steering vector by averaging the computed differences between $\mathbf{z}_{qms}^{+}$ and $\mathbf{z}$ across the image set $\mathbf{X}$, which is a common practice for activation editing

(Chen et al., 2025; Li et al., 2024b):

$$\mathbf{d}_{\text{qms}} = \frac{1}{|\mathbf{X}|} \sum_{\mathbf{X}} (\mathbf{z}_{\text{qms}}^{+} - \mathbf{z}), \tag{5}$$

where $\mathbf{d}_{\text{qms}}$ denotes the manifestation-aware editing vector, $|\mathbf{X}|$ denotes the number of calculated image samples. Therefore, QMS can activate the manifestation expertise in $\mathcal{M}$ by applying $\mathbf{d}_{\text{qms}}$ to perform beneficial editing, thereby enhancing the perception of medical imaging.

### 3.3. Principle-driven Diagnosis Induction

Additionally, we propose Principle-driven Diagnosis Induction (PDI) to address the limitations of previous methods, which either fail to capture clinician expertise (Liu et al., 2024b) or depend on costly fine-grained region annotations (Li et al., 2025a). Motivated by the textbook-recommended principles (Goodman, 2014; Klein & Rosado-de Christenson, 2019), PDI accordingly designs a scope-conditioned template that explicitly embeds these expert diagnostic principles, and instantiates the template with decomposed clinical reports to construct positive diagnostic guidance. Therefore, PDI derives principle-aware vectors to steer Med-LVLMs toward expert-like diagnostic cognition.

**Scope-conditioned Diagnostic Template** Grounded in textbook-recommended principles (Goodman, 2014) that advocate a global-to-local reading strategy with interpretation in a prescribed order (*e.g.*, the classical ABCDE approach (Klein & Rosado-de Christenson, 2019) for chest radiographs), PDI devises a scope-conditioned diagnostic template $\mathbf{P} = \|_{i=1}^{3} p_i$ to explicitly encode these principles, where $\|$ denotes concatenation operator. Specifically, $\mathbf{P}$ comprises three textual sub-templates with fillable slots, which are deliberately scoped to operationalize the diagnostic workflow in a global $\rightarrow$ local $\rightarrow$ summary manner. Notably, $p_2$ further encodes the necessary order like ABCDE approach, thereby enforcing principle-consistent evidence organization. The sub-templates are shown as follows:

- **Global Assessment** $p_1(a)$: A coarse and holistic overview of the entire image, with slot $[a]$ to be populated with global findings or study-level observations:

    *Firstly, the global observations and findings of this medical image are:* $[a]$.

- **Local evidence** $p_2(a, b)$: A sorted collection of focused local regions and evidence according to the recommended principles, with slot [Sort($a$)] filled with focused regions and slot [Sort($b$)] filled with findings:

    *Based on the query, the following regions should be focused:* [Sort($a$)].
    *According to the focused region, the key findings are:* [Sort($b$)].

- **Diagnostic Statement** $p_3(a)$: A diagnostic summary, with slot $[a]$ filled by overall impressions:

    *In summary, the patient was diagnosed with the following impressions:* $[a]$.

**Scope-aligned Template Instantiating** Subsequently, we can restructure existing reports through scope-aware decomposing and instantiate their contents into $\mathcal{P}$, yielding a chain-structured diagnostic prompt. Specifically, given a collected report $\mathbf{T} = \{t_1, ..., t_n\}$ for each image $x$ in $\mathbf{X}$, where $t_j$ denotes an individual sentence, PDI queries $\mathcal{F}$ with an instruction $\mathbf{I}_{\text{dec}}$ to infer its scope label $y_j \in \{1, 2, 3\}$, corresponding to the global, local, and summary scope levels of the template $\mathbf{P}$, respectively. Meanwhile, $\mathcal{F}$ also extracts an anatomical region $r_j$, which is defined only for localized sentences, and set to `None` for the remaining scopes. Finally, based on the inferred scope label $y_j$, each sentence $t_j$ and anatomical region $r_j$ is populated into the corresponding scope-specific sub-template in $\mathbf{P}$, thereby instantiating the template and yielding a reasoning-chain prompt $w$ that embeds the expert interpretation principles:

$$w = \|_{i=1}^{3} \text{Fill}(p_i; \{(r_j, t_j) | y_j = i\}), \tag{6}$$

where $\text{Fill}(\cdot; \cdot)$ represents a slot-filling operation. It is noticed that $r_j$ is only instantiated for $p_2$ and ignored for the remaining scopes.

**Principle-aware Vector Steering** Analogously to QMS, PDI derives the principle-aware steering vector $\mathbf{d}_{\text{pdi}}$ by contrasting the trusted activation $\mathbf{z}_{\text{pdi}}^{+}$ of chain-augmented sample $(x, w, q)$ with the untrusted activation $\mathbf{z}$ of original input $(x, q)$, thereby improving the imaging cognition of Med-LVLM:

$$\mathbf{d}_{\text{pdi}} = \frac{1}{|\mathbf{X}|} \sum_{\mathbf{X}} (\mathbf{z}_{\text{pdi}}^{+} - \mathbf{z}). \tag{7}$$

### 3.4. Expertise-informed Cooperative Steering

Finally, MEDA performs cooperative steering by jointly applying the vectors learned from QMS and PDI, thereby encouraging comprehensive utilization of medical expertise. Concretely, following (Wang et al., 2026), we rank all head-wise vectors $\{\mathbf{d}_{\text{qms}}^{(l,h)}\}$ and $\{\mathbf{d}_{\text{pdi}}^{(l,h)}\}$ by their $l_2$-norms and conduct editing using the top-$K$ vectors:

$$\begin{aligned}
\hat{\mathbf{z}}^{(l,h)} = \mathbf{z}^{(l,h)} + \alpha \cdot (&\mathbb{I}[(l, h) \in \mathbf{H}_{\text{qms}}] \cdot \mathbf{d}_{\text{qms}}^{(l,h)} \\
&+ \mathbb{I}[(l, h) \in \mathbf{H}_{\text{pdi}}] \cdot \mathbf{d}_{\text{pdi}}^{(l,h)}),
\end{aligned} \tag{8}$$

where $\mathbf{H}_{\text{qms}}$ and $\mathbf{H}_{\text{pdi}}$ denote the edited head sets for QMS and PDI, respectively, such that $|\mathbf{H}_{\text{qms}}| + |\mathbf{H}_{\text{pdi}}| = K$.

| Methods | IU-Xray | | | | | MIMIC-CXR | | | | | VQA-RAD | | SLAKE | |
|---|---|---|---|---|---|---|---|---|---|---|---|---|---|---|
| | ACC | F1 | MT | RG | RS | ACC | F1 | MT | RG | RS | ACC | F1 | ACC | F1 |
| **Baseline** | 74.2 | 69.9 | 24.8 | 22.0 | 50.8 | 72.5 | 75.4 | 20.6 | 25.3 | 39.0 | 58.6 | 60.2 | 53.8 | 55.7 |
| *Training-based & RAG-based Methods* | | | | | | | | | | | | | | |
| **RULE** | 80.9 | 76.1 | 25.0 | 24.6 | 50.5 | 76.7 | 77.9 | 23.1 | 26.9 | 39.3 | 60.9 | 63.7 | 54.5 | 57.0 |
| **MMedRAG** | 81.1 | 75.8 | 24.0 | 24.9 | 50.4 | 77.2 | 79.2 | 22.4 | 27.1 | 38.9 | 62.5 | 63.8 | 54.5 | 56.6 |
| **MMedPO** | 79.1 | 74.0 | 26.4 | 24.7 | 56.7 | 76.6 | 78.2 | 23.5 | 27.6 | 38.9 | 63.2 | **64.2** | 55.7 | 57.4 |
| **ExGra-Med** | 81.7 | 76.3 | 23.4 | 25.6 | 51.3 | 74.5 | 76.8 | 20.1 | 25.6 | 39.8 | 60.7 | 60.1 | 54.6 | 54.5 |
| *Inference-time Activation Editing Methods* | | | | | | | | | | | | | | |
| **VTI** | 78.8 | 75.8 | 22.1 | 19.8 | 48.6 | 72.8 | 76.3 | 19.2 | 24.4 | 40.6 | 58.8 | 62.2 | 52.6 | 53.6 |
| **ICT** | 79.9 | 76.0 | 25.0 | 22.6 | 51.9 | 74.6 | 76.6 | 20.8 | 26.6 | 40.3 | 60.6 | 62.1 | 54.3 | 54.9 |
| **VISTA** | 76.4 | 73.1 | 24.4 | 23.4 | 54.2 | 74.2 | 78.1 | 23.4 | 26.2 | 39.7 | 58.0 | 61.9 | 55.7 | 58.8 |
| **AFTER** | 81.4 | 76.9 | 26.8 | 24.6 | 53.6 | 77.2 | 78.1 | 21.8 | 25.9 | 40.4 | 62.1 | 63.0 | 55.3 | 56.7 |
| **Ours** | **84.8** | **79.7** | **30.2** | **27.8** | **59.2** | **79.4** | **79.7** | **24.1** | **30.5** | **42.1** | **63.4** | 63.5 | **58.1** | **59.0** |

*Table 1.* Comparison of MEDA with SOTA methods on LLaVA-Med-1.5 across IU-Xray, MIMIC-CXR, VQA-RAD, SLAKE. The best results are in **bold**. Each result is reported under multiple rounds. The short names "MT", "RG", "RS" refer to METEOR, ROUGE-1, and RaTEScore metrics for report generation, respectively.

## 4. Experiments

In this section, we conduct comprehensive experiments across multiple datasets, modalities and LVLMs to illustrate the effectiveness and practical applicability of MEDA.

### 4.1. Experimental Setup

**Benchmarks and Metrics** We employ six widely-adopted medical vision-language datasets for medical VQA and report generation tasks across multiple clinical modalities, including radiology (IU-Xray (Demner-Fushman et al., 2015), MIMIC-CXR (Johnson et al., 2019), VQA-RAD (Lau et al., 2018), SLAKE (Liu et al., 2021)), ophthalmology (Harvard-FairVLMed (Luo et al., 2024)), and pathology (PMC-OA (Lin et al., 2023)). Specifically, we adopt the MedHEval benchmark (Chang et al., 2025) question sets for hallucination evaluation on the four radiology datasets. For Harvard-FairVLMed and PMC-OA, we directly use the test samples provided in (Xia et al., 2025). Following (Chang et al., 2025; Xia et al., 2025), we evaluate VQA performance using Accuracy and F1-score, and assess report generation quality using METEOR (Banerjee & Lavie, 2005), ROUGE-1 (Lin, 2004), and RaTEScore (Zhao et al., 2024). Additional details are provided in Appendix B.

**Baseline and Comparative Methods** We choose six advanced LVLMs, including four Med-LVLMs (LLaVA-Med-1.5 (Li et al., 2023), CheXagent (Chen et al., 2024b), LLM-CXR (Lee et al., 2024), and LLaVA-Tri (Xie et al., 2025)) and two generic LVLMs (Qwen2.5-VL-Instruct (Bai et al., 2025) and LLaVA-Next (Liu et al., 2024a)) as our baselines, with a particular focus on classic LLaVA-Med-1.5 for primary experiments. To evaluate our superiority, we first compare MEDA with generic activation editing methods, *i.e.*

| Methods | Harvard-FairVLMed | | PMC-OA | |
|---|---|---|---|---|
| | ACC | RS | ACC | RS |
| **Baseline** | 85.6 | 30.3 | 60.4 | 29.9 |
| **VTI** | 85.4 | 30.4 | 62.5 | 22.1 |
| **ICT** | 87.3 | 30.4 | 63.0 | 30.0 |
| **VISTA** | 87.8 | 30.9 | 63.2 | 29.6 |
| **AFTER** | 87.3 | **32.4** | 64.1 | 30.1 |
| **Ours** | **88.3** | 32.1 | **65.5** | **34.2** |

*Table 2.* Comparison of MEDA with generic activation editing methods on Harvard-FairVLMed and PMC-OA.

VTI (Liu et al., 2025), ICT (Chen et al., 2025), VISTA (Li et al., 2025b), and AFTER (Wang et al., 2026). We also consider SOTA medical RAG methods RULE (Xia et al., 2024) and MMed-RAG (Xia et al., 2025), and training-based methods MMedPO (Zhu et al., 2025a) and ExGra-Med (Nguyen et al., 2025) tailored for medical scenario.

**Implementation Details** For vector construction, we use the close-ended "Visual Misinterpretation" subset of MIMIC-CXR for radiology, and the first 1,000 samples from the VQA sets of Harvard-FairVLMed and PMC-OA for ophthalmology and pathology, respectively. We strictly exclude all construction data from the test sets to prevent data leakage, and evaluate all baselines on the same test splits to ensure fair comparison. We default to GPT-4 as $\mathcal{F}$, with Radiopaedia (Gaillard et al., 2011), PathologyOutlines (PathologyOutlines.com, Inc., 2026), and EyeWiki (American Academy of Ophthalmology, 2026) serving as the database $\mathcal{D}$. Without specifying, we set the threshold $\tau$ to 0.8, the edited head num $K$ to 32, and the editing strength $\alpha$ to 5. More details are provided in Appendix C.

## 4.2. Experimental Results

**Overall Performance**   Table 1 shows the comparison between MEDA and various hallucination mitigation methods on LLaVA-Med-1.5 across four radiology datasets. Obviously, our method demonstrates significant advantages across all datasets in both medical VQA and report generation tasks. We particularly achieve an improvement of 10.6% in IU-Xray accuracy and 6.9% in MIMIC-CXR accuracy over the baselines, surpassing the SOTA editing method AFTER by 3.4% and 2.2%. Additionally, on the report generation task of IU-Xray, MEDA yields averaged improvements of 6.5% across three generative metrics, outperforming all SOTA methods. This enhancement demonstrates the superiority of the positive imaging manifestation and diagnostic principle, which provides more clinically grounded guidance than coarse visual contrast, thereby effectively mitigating medical hallucination. Results on Harvard-FairVLMed and PMC-OA in Table 2 further demonstrate consistent performance gains on both tasks, highlighting the broad applicability of MEDA across diverse medical imaging modalities. Besides the classic LLaVA-Med-1.5, we further validate five additional sophisticated LVLMs with varying architectures and parameter sizes. Figure 3 shows that we can effectively enhance the truthfulness of all models on MIMIC-CXR, yielding average improvements of 5.9% in accuracy and 5.4% in RaTEScore.

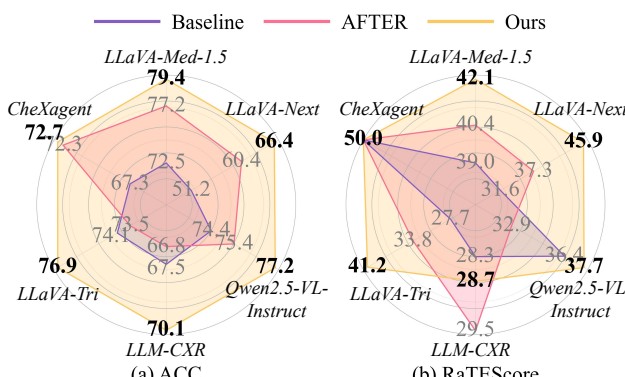

*Figure 3.* Performance of our MEDA across various advanced LVLMs on MIMIC-CXR.

**Cross-dataset Generalization**   Notably, the results on IU-Xray, VQA-RAD, and SLAKE reported in Table 1 are obtained by generalizing the steering vectors learned on MIMIC-CXR to the corresponding target datasets. Despite differences in image sources, MEDA consistently delivers notable performance improvements across these datasets. This indicates that MEDA can achieve general medical hallucination mitigation of Med-LVLMs rather than merely fitting a specific dataset, therefore exhibiting strong potential for generalization to open-world scenarios.

| QMS | PDI | ACC | F1 | RS |
|:---:|:---:|:---:|:---:|:---:|
| **Baseline** | | 74.2 | 69.9 | 50.8 |
| ✓ | | 81.2(↑**7.0%**) | 75.8(↑**5.9%**) | 56.6(↑**5.8%**) |
| | ✓ | 82.5(↑**8.3%**) | 77.1(↑**7.2%**) | 57.9(↑**7.1%**) |
| ✓ | ✓ | **84.8**(↑**10.6%**) | **79.3**(↑**9.4%**) | **59.2**(↑**8.4%**) |

*Table 3.* The ablation study of MEDA on IU-Xray.

## 4.3. Ablation Study

**Ablation of MEDA**   We first investigate QMS and PDI modules respectively in Table 3. The baseline model without editing achieves 74.2% on the primary metric Accuracy. Simply employing the QMS module will bring huge 7.0% performance gains. It reveals that enhancing the activations with the positive imaging manifestation guidance indeed increases the factual perception of medical images, thus benefiting hallucination mitigation. Additionally, solely deploying the PDI module will bring 8.3% improvement. It manifests that editing with the guidance of positive diagnostic principles is also essential for improving truthfulness. However, without cooperative steering, the Med-LVLM cannot reach peak performance. Therefore, the QMS and PDI module should mutually enhance their functionalities.

**Ablation of Hyperparameters**   We first analyze two hyperparameters that regulate the editing, *i.e.* the number of edited heads $K$ and editing strength $\alpha$. From Figure 4(a), we can observe that the accuracy exhibits an inverted U-shaped curve. The best accuracy (84.8%) is achieved at $K = 32$, $\alpha = 5$, which demonstrates the effectiveness of editing with appropriately calibrated editing strength. The declines under excessive steering reveal a trade-off between truthfulness and helpfulness for editing methods (Li et al., 2024a; Chen et al., 2025), providing us with intuitive guidance for editing. We further vary the threshold $\tau$, which is used to assess both category relevance during entity extraction and the entity-term similarity during retrieval, with the results reported in Figure 4(b). We observe that the accuracy reaches its optimum when $\tau \geq 0.5$, where the retrieved manifestations are relatively precise. As $\tau$ decreases, performance degrades due to the priority mechanism overly dominating entity generation, leading to disease entities being produced despite limited query-specific relevance.

## 4.4. In-Depth Analysis

**Analysis of Model $\mathcal{F}$**   To investigate the impact of the capability of $\mathcal{F}$ on MEDA, we evaluate three different models with varying architectures (closed-source GPT-4o *vs.* open-source Qwen2.5-VL-Instruct) and model scales (200B *vs.* 8B). As shown in Figure 5(a), there is minimal performance variation between the entities and scope labels generated by the three models $\mathcal{F}$. This robustness can be

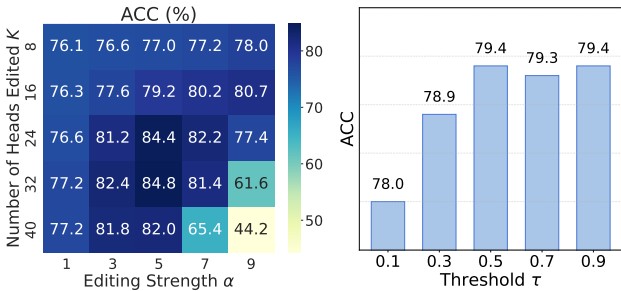

*Figure 4.* Ablation of edited heads $K$ and $\alpha$ on IU-Xray (a, left), and threshold $\tau$ on MIMIC-CXR(b, right).

attributed to the category-aware extraction mechanism in QMS, which guides $\mathcal{F}$ with category-priority priors to mitigate entity extraction errors arising from limited model capacity. In addition, the diagnostic template $\mathbf{P}$ explicitly encodes textbook-recommended principles, reducing the sensitivity of MEDA to discrepancies in scope inference. Collectively, this robustness supports the low-cost deployment of MEDA in real-world clinical settings.

**Analysis of Few-shot setting** To explore MEDA's sensitivity to training data availability, we vary the sample size used to construct vectors from 10 to 1300, including few-shot settings of 10, 30, and 50 training samples. As shown in Figure 5(b), MEDA consistently achieves strong performance across different scales on all metrics, with primary accuracy steadily rising from 78.2% to 79.4%. Notably, even with only 10 samples, MEDA attains ACC, F1, and RaTEScore of 78.2%, 78.9%, and 41.1%, outperforming the baseline LLaVA-Med-1.5 by 5.7%, 3.5%, and 2.1%. This is credited to the effective incorporation of imaging manifestation and diagnostic principles, highlighting the practical advantage of MEDA in low-data scenarios.

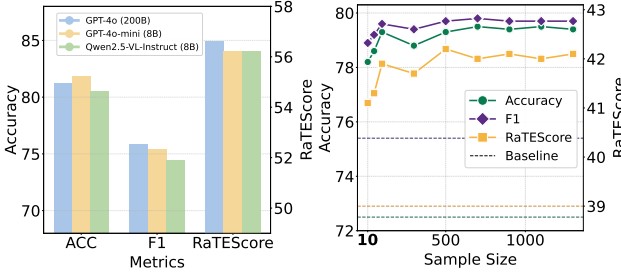

*Figure 5.* Detailed analysis of various model $\mathcal{F}$ on IU-Xray (a, left) and sample size on MIMIC-CXR (b, right).

**Analysis of Efficiency** We also compare MEDA with SOTA methods to illustrate its efficiency in both training (vector construction) and inference. All experiments are conducted on 80G A800s for fair comparison. Figure 6(a) first compares the training resource consumption of two representative methods that require fine-tuning. Notably, MEDA

completes vector construction with the smallest amount of training data in 0.5 GPU-hours, achieving a 34.1× speedup over MMedRAG, which requires 17.1 GPU-hours. Additionally, we compare inference speed and hallucination mitigation results on MIMIC-CXR against other inference-time methods. Results in Figure 6(b) demonstrate that MEDA achieves the best mitigation performance while maintaining the fastest inference speed of 0.48 samples per second. Apart from the inference speed, MEDA also maintains moderate memory usage of 15.9 GB (expressed as the volume of spheres), facilitating practical deployment without demanding excessive resources.

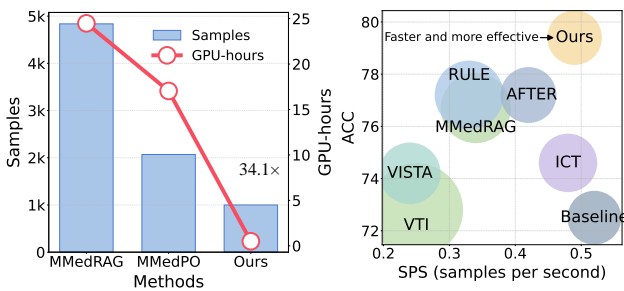

*Figure 6.* Comparison of training (vector construction) cost (a, left) and inference cost (b, right).

**Case Study** In Figure 7, we present two representative case studies of LLaVA-Med-1.5 on the IU-Xray and PMC-OA. In the first radiology case, LLaVA-Med-1.5 hallucinates a right-sided pleural effusion despite the absence of supporting visual evidence. In contrast, guided by positive imaging manifestations, MEDA produces a clinically accurate report indicating that the lungs are clear. MEDA avoids the hallucinated prediction in the second pathology case, demonstrating appropriate medical image interpretation.

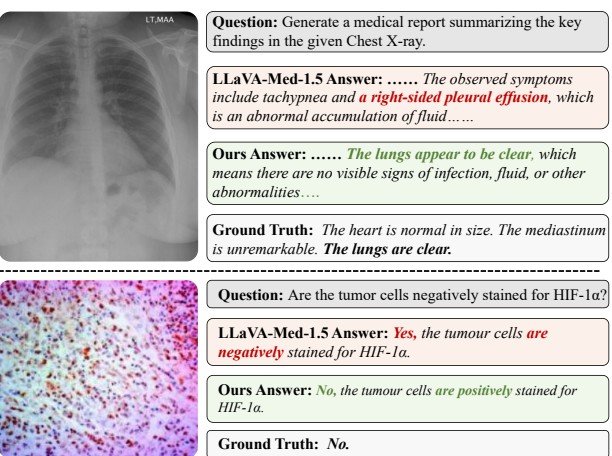

*Figure 7.* Two case studies on IU-Xray and PMC-OA.

# 5. Conclusion

In this paper, we propose MEDA, the first medical-oriented activation editing method that steers the Med-LVLMs toward positive imaging manifestation and diagnostic principle guidance for mitigating hallucination. Extensive experiments have confirmed that MEDA achieves superior mitigation performance with minimal cost, while exhibiting cross-dataset generalization and few-shot robustness for clinical applications. One limitation of MEDA, as well as other activation editing methods, is its reliance on access to model activations, which restricts its applicability to closed-source LVLMs. In future work, we intend to extend MEDA to encompass other specialized domains.

# Impact Statement

This paper aims to improve the factual reliability of Medical LVLMs by mitigating hallucinations through medically grounded activation editing. By explicitly steering models with imaging manifestations and diagnostic principles, the proposed method has the potential to enhance the safety and trustworthiness of medical AI systems used in research, education, and clinical decision support. Improved hallucination mitigation may help reduce misleading or unsupported model outputs, which is particularly important in high-stakes medical contexts.

To further ensure responsible use, MEDA relies on curated and authoritative medical sources and structured diagnostic templates, which help maintain semantic alignment with established clinical knowledge. In practice, maintaining diversity, quality, and periodic updates of the underlying medical references can further strengthen the reliability of the induced guidance. We also encourage domain-aware validation when applying MEDA in specialized medical contexts, ensuring that the steering process remains consistent with current clinical standards. With appropriate oversight, MEDA provides a robust and principled framework for enhancing the factual integrity of Medical LVLMs.

# Acknowledgements

This work was supported by the National Key Research and Development Program of China (NO. 2023YFC2506800), the Young Elite Scientists Sponsorship Program of the Beijing High Innovation Plan (NO. 20250627).

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

## Appendix Content

## A. More Experimental Analysis

### A.1. Deep Ablation of QMS

In Table 4, we first investigate the contributions of key components in QMS. The results show that removing threshold control for manifestation retrieval slightly degrades performance by 1.1% accuracy, 1.3% RaTEScore for IU-Xray and 0.9% accuracy, 0.5% RaTEScore for MIMIC-CXR, indicating that validity filtering is necessary to avoid injecting mismatched or low-confidence knowledge when the extracted entity has no reliable correspondence in the database. Moreover, removing the priority weight for entity extraction causes a larger drop of 2.4% accuracy, 2.2% RaTEScore for IU-Xray and 1.2% accuracy, 0.8% RaTEScore for MIMIC-CXR, demonstrating that category-aware prioritization helps identify the most diagnostically decisive entity, thereby improving the relevance of retrieved manifestations and the effectiveness of subsequent steering. Finally, removing both components leads to the worst ablated performance, suggesting that priority-guided extraction and threshold-controlled retrieval are complementary: the former improves what to retrieve, while the latter ensures when retrieval is reliable, together yielding the most stable and effective manifestation guidance.

### A.2. Deep Ablation of PDI

Table 4 also reports the ablation study of PDI, quantifying the contribution of its principle encoding mechanisms. Specifically, removing the prescribed-order sort in the local-evidence sub-template leads to a small but consistent drop, with accuracy and RaTEScore decreasing from 82.5% to 81.9% and from 57.9% to 57.0% on IU-Xray. This indicates that enforcing a clinician-recommended reading order (e.g., ABCDE-style organization) helps structure localized evidence in a way that better supports subsequent diagnosis formation, even when the same evidence is provided. In contrast, removing the scope-conditioned diagnostic template causes a substantially larger degradation, with IU-Xray dropping to 80.2% accuracy and 55.2% RaTEScore. This result highlights the central role of the template design: merely reusing the original report does not sufficiently elicit expert-like diagnostic cognition, whereas explicit scope separation provides a stronger inductive bias

for principle-consistent reasoning. Finally, using only the raw report as positive guidance (removing both sorting and the template) yields the weakest ablated configuration. This suggests that PDI's gains are not attributable to additional text alone; rather, they stem from principle-driven organization of evidence and impressions, with the template serving as the primary driver and prescribed-order sorting providing complementary improvements.

| Ablation Configuration | IU-Xray ACC | RS | MIMIC-CXR ACC | RS |
|---|---|---|---|---|
| **Baseline** | 74.2 | 50.8 | 72.5 | 39.0 |
| **Full QMS** | **81.2** | **56.6** | **78.1** | **41.4** |
| *- Threshold Control for Manifestation Retrieval* | 80.1 | 55.3 | 77.2 | 40.9 |
| *- Priority Weight for Entity Extraction* | 78.8 | 54.4 | 76.9 | 40.6 |
| *- Both Threshold Control and Priority Weight* | 77.9 | 53.8 | 76.6 | 40.1 |
| **Full PDI** | **82.5** | **57.9** | **78.9** | **42.3** |
| *- Prescribed Order Sort* | 81.9 | 57.0 | 78.0 | 42.0 |
| *- Scope-conditioned Diagnostic Template* | 80.2 | 55.2 | 77.1 | 40.3 |
| *- Both Sort and Template (only report)* | 79.8 | 54.9 | 77.0 | 39.7 |

*Table 4.* Deep ablations of QMS and PDI on two datasets. The notation "–" denotes the removal of the corresponding component from its associated module.

### A.3. Analysis of Magnitude Distribution

To examine why Med-LVLMs struggle to properly leverage internal knowledge and activate clinically grounded reasoning within their architecture, we analyze the magnitude distributions of the manifestation-aware and principle-aware editing vectors across all layers and attention heads in Figure 8. For both vectors, we observe a pronounced rise in magnitudes in the middle layers (layers 9–17), suggesting that the model's internal representations undergo the most intensive knowledge- and principle-related adjustment at this stage, where multimodal features are being integrated. This pattern indicates that, even when relevant cues are present, the model does not reliably align intermediate visual representations with clinically meaningful manifestations, nor does it consistently invoke diagnostic principles to constrain interpretation. Consequently, the mismatch between what is visually supported and what is selected and verbalized persists and compounds through later layers, eventually surfacing in the final decoding layers as unsupported or overconfident statements.

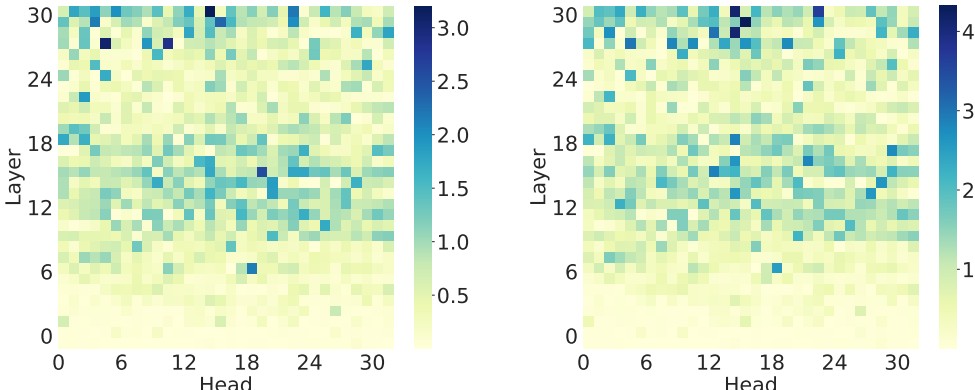

*Figure 8.* Distribution of magnitudes for manifestation-aware vectors derived by QMS (a, left figure) and principle-aware vectors derived by PDI (b, right figure).

## B. More Details on Datasets

### B.1. Radiology Datasets: MIMIC-CXR, IU-Xray, RAD-VQA, SLAKE

The radiology datasets used in our experiments, *i.e.* MIMIC-CXR, IU-Xray, VQA-RAD, and SLAKE, are widely adopted benchmarks that have been extensively used in prior studies (Chang et al., 2025; Xia et al., 2024; 2025). In our experiments, we directly adopt the MedHEval benchmark (Chang et al., 2025), which is specifically designed to systematically evaluate

hallucinations and corresponding mitigation strategies in Med-LVLMs by categorizing them into three underlying causes: visual misinterpretation, knowledge deficiency, and context misalignment. Specifically, MedHEval constructs close-ended Visual Misinterpretation test sets with more than 1,500 samples for each of the four radiology datasets, while the close-ended Knowledge Deficiency and Context Misalignment subsets are provided only for MIMIC-CXR. In addition, MedHEval offers open-ended report generation benchmarks for IU-Xray and MIMIC-CXR. In practice, we use the close-ended Visual Misinterpretation subset of MIMIC-CXR exclusively for vector construction, and reserve all other subsets for evaluation to strictly avoid data leakage. Consequently, the close-ended MIMIC-CXR results reported in Table 1 are averaged over the Knowledge Deficiency and Context Misalignment subsets. Analogous to the procedure used to extract pathology-related questions from PMC-OA in (Xia et al., 2025), we further restrict the RAD-VQA and SLAKE benchmarks to X-ray modality questions from their validation splits.

### B.2. Ophthalmology Dataset: Harvard-FairVLMed

Following (Xia et al., 2025), we use Harvard-FairVLMed, which contains multimodal fundus images collected from diverse sources, to evaluate the applicability of MEDA in ophthalmology. Specifically, (Xia et al., 2025) constructed a VQA benchmark for Harvard-FairVLMed by generating question–answer pairs from medical reports using GPT-4, with answers formatted in a binary (yes/no) manner. In our experiments, we use the first 1,000 samples from the constructed VQA set for vector construction, and reserve the remaining 3,285 samples as the test set to ensure a strict separation between construction and evaluation data. For the report generation task, we directly use the test set provided in (Xia et al., 2025).

### B.3. Pathology Dataset: PMC-OA

Consistent with the protocol used for Harvard-FairVLMed, we adopt the PMC-OA VQA dataset constructed in (Xia et al., 2025) as the benchmark for vector construction and evaluation. Specifically, we use the first 1,000 samples from the constructed VQA set for vector construction and reserve the remaining 1,155 samples as the test set. For the report generation task, we directly use the test set provided in (Xia et al., 2025).

## C. More details on Implementation

### C.1. Database Implementation

The retrieval databases used in our QMS module are derived from Radiopaedia (Gaillard et al., 2011), PathologyOutlines (PathologyOutlines.com, Inc., 2026), and EyeWiki (American Academy of Ophthalmology, 2026). To mitigate the practical overhead of online querying during large-scale experiments, we construct a lightweight local database in a dictionary-style format. Taking Radiopaedia as an example, we first submit all entities extracted from the questions to the website's search engine, and then crawl the first-page articles by collecting titles along with their associated manifestation descriptions, including the introductory paragraph and radiologic features section. The resulting entries are indexed and stored locally. Retrieval is subsequently performed following the procedure described in Section 3.2. This implementation preserves the retrieval content and process while substantially improving experimental efficiency without affecting performance.

### C.2. Hyperparameter Setting

Following (Chen et al., 2025; Wang et al., 2026), we use hyperparameter tuning to determine editing strength $\alpha$ and edited heads num $K$ and ensure reproducibility. It is noticed that due to the uninterpretable nature of LVLMs, existing activation editing methods cannot specify the optimal editing strength or the appropriate head number without an established theory, and therefore rely on empirical hyperparameter tuning. Hyperparameter tuning was performed using a grid search to explore the possible combinations of key parameters systematically. Specifically, the search space was defined as the Cartesian product: $\{1, 3, 5, 7, 9\} \times \{8, 16, 24, 32, 40\}$, where the first discrete set denotes the selected values of $\alpha$, which range was chosen to strike a balance between improving model trustworthiness and maintaining overall performance, and the second is for $K$, which allows for sparse editing (total 1024 heads) for both effectiveness and efficiency.

Consistent with the observation in (Wang et al., 2026), MEDA can maintain stable performance on new datasets when initialized with the same prior hyperparameters. Only minor adjustments are typically required to reach optimal performance.

| Methods | Knowledge Deficiency | | Context Misalignment | |
|---|---|---|---|---|
| | ACC | F1 | ACC | F1 |
| **Baseline** | 64.6 | 69.1 | 80.5 | 83.8 |
| *Training-based & RAG-based Methods* | | | | |
| **RULE** | 71.3 | 71.9 | 82.1 | 83.9 |
| **MMedRAG** | 71.2 | **72.6** | 83.3 | 85.8 |
| **MMedPO** | 70.4 | 71.0 | 82.8 | 85.4 |
| **ExGra-Med** | 68.8 | 69.3 | 80.2 | 84.4 |
| *Inference-time Activation Editing Methods* | | | | |
| **VTI** | 63.4 | 69.6 | 82.2 | 85.0 |
| **ICT** | 67.6 | 69.6 | 81.6 | 83.6 |
| **VISTA** | 67.6 | 71.4 | 80.7 | 84.8 |
| **AFTER** | 71.0 | 70.4 | 83.5 | 85.8 |
| **Ours** | **72.3** | 71.3 | **86.5** | **88.1** |

*Table 5.* Full experimental results of close-ended "Knowledge Deficiency" and "Context Misalignment" subsets of MIMIC-CXR.

## D. Full Experimental Results

As illustrated in Appendix B.1, the Accuracy and F1-score of MIMIC-CXR reported in Table 1 are the averaged results of close-ended "Knowledge Deficiency" and "Context Misalignment" subsets of MIMIC-CXR in MedHEval. Therefore, we present the full experimental results of close-ended "Knowledge Deficiency" and "Context Misalignment" in Table 5.

## E. More Details on Prompts

The prompt $I_{sco}$ used to assess the relevance score of query-answer pair $(q, a)$ to each category $c$ is as follows:

> Your task is to assess the relevance between the given question–answer pair and the specified category, and assign a numerical score between 0 and 1. Please only return the relevance score.
>
> Question: [Question]
> Answer: [Answer]
> Category: [Category]
>
> Score:

The prompt $I_{ent}$ used to extract the entity from the question-answer pair according to the selected category is as follows:

> Your task is to extract one most important entity from a given question and answer that belongs to the specified category. By acquiring more knowledge about this entity, the model can better answer the question. Please only return the entity name. Additionally, if the question is not related to any entity (e.g. the question: What is the modality of the image?), you should return "None".
>
> Question: [Question]
> Answer: [Answer]
> Category: [Category]
>
> Entity:

The prompt $I_{dec}$ used to infer the scope label and extract the anatomical region for each sentence is as follows:

Your task is to analyze a given sentence from a medical report and determine its diagnostic scope. The scope must be one of the following three types: (1) Global assessment: a holistic or study-level observation of the entire image; (2) Local evidence: a localized finding referring to specific anatomical regions; (3) Diagnostic summary: an overall diagnostic impression or conclusion.
Please assign a scope label according to the definitions above, where 1 = Global assessment, 2 = Local evidence, and 3 = Diagnostic summary.

In addition, if the sentence is classified as Local evidence (scope = 2), please extract the corresponding anatomical region mentioned in the sentence. If the sentence does not describe a localized finding, return "None" for the anatomical region.
Please only return the scope label and the anatomical region in the specified format.

Sentence: [Sentence]

Scope label:
Anatomical region:

