# OpenReview forum: "MEDA: Medical-Oriented Activation Editing for Hallucination Mitigation in Medical Large Vision-Language Model"
_ICML.cc/2026/Conference — ICML 2026 regular_

### Official Review · Reviewer_YjJS · 2026-03-10

**Soundness:** 2
**Presentation:** 3
**Significance:** 2
**Originality:** 3
**Overall Recommendation:** 4
**Confidence:** 3

**Summary:**

This paper introduces MEDA, an efficient, inference-time activation editing framework designed to mitigate hallucinations in Medical Large Vision-Language Models (Med-LVLMs) at a minimal computational cost. By explicitly retrieving professional imaging manifestations and incorporating textbook-level diagnostic principles, MEDA directly steers the model's internal activation states toward clinically accurate reasoning.

**Compliance With Llm Reviewing Policy:**

Affirmed.

**Final Justification:**

Strengths: The paper provides a highly efficient and effective solution for hallucination mitigation in medical AI. Its primary strengths are its SOTA empirical performance and its impressive resource efficiency, requiring minimal samples and GPU time.

Weaknesses: The main limitation is the lack of a rigorous theoretical foundation. While the method works well, the explanation for why these specific clinical constraints lead to linear separability in the activation space remains largely based on heuristics rather than formal proof.

Impact of Rebuttal: The authors' response addressed my concerns without changing my prior assessment. I maintain my recommendation of weak accept.

**Key Questions For Authors:**

1. While the empirical performance of QMS and PDI is impressive, the paper would be significantly strengthened by a theoretical discussion. How do the authors theoretically explain the linear separability of such complex, multi-step clinical reasoning within the model's intermediate activation layers?

2. The current methodology brilliantly exploits the structured heuristics of clinical diagnostics. However, to better understand the broader impact of this work, could the authors discuss its generalizability? Specifically, what adaptations would be necessary to apply this activation editing paradigm to non-medical domains where expert reasoning is less rigidly templated?

**Limitations:**

yes

**Strengths And Weaknesses:**

1. MEDA achieves state-of-the-art hallucination mitigation across multiple medical datasets, significantly improving factuality.

2. Furthermore, it boasts extremely low training and deployment costs, requiring as few as 10 samples and only 0.5 GPU hours to construct its highly effective steering vectors.

3. The proposed MEDA framework heavily relies on empirical clinical heuristics (e.g., the ABCDE diagnostic principle) rather than providing a rigorous theoretical foundation for why these specific constraints form linearly separable vectors in the activation space.

---

> ### Author Rebuttal · Authors · 2026-03-31
>
> We are truly thankful for your comprehensive review and the appreciation of our effectiveness
>
> > ***W3:** The proposed MEDA framework heavily relies on empirical clinical heuristics (e.g., the ABCDE diagnostic principle) rather than providing a rigorous theoretical foundation for why these specific constraints form linearly separable vectors in the activation space.*
>
> > ***Q1:** While the empirical performance of QMS and PDI is impressive, the paper would be significantly strengthened by a theoretical discussion. How do the authors theoretically explain the linear separability of such complex, multi-step clinical reasoning within the model's intermediate activation layers?*
>
> Thank you for providing such constructive feedback. Drawing on prior studies[1,2], our core hypothesis is that **the Med-LVLM forms behavior-relevant (truthful versus hallucinated behavior) hidden states in certain intermediate activation layers that may be linearly separable**. In the context of MEDA, hidden states that more effectively integrate multi-step clinical reasoning are more likely to lead to **truthful** behavior, whereas the original hidden states are more prone to produce **hallucinated** behavior. Therefore, truthful versus hallucinated behavior can be operationalized as linear direction in the activation space and intervened upon accordingly[2]. We welcome further theoretical investigation of this perspective.
>
> > ***Q2:** The current methodology brilliantly exploits the structured heuristics of clinical diagnostics. However, to better understand the broader impact of this work, could the authors discuss its generalizability? Specifically, what adaptations would be necessary to apply this activation editing paradigm to non-medical domains where expert reasoning is less rigidly templated?*
>
> Thank you for your meaningful comment. For **specialized non-medical domains with well-defined reasoning principles** (e.g. scientific tasks in remote sensing), MEDA can **naturally generalize by replacing clinical diagnostic principles with domain-specific ones**. To provide preliminary evidence, we evaluate MEDA on 100 randomly sampled questions from the remote sensing benchmark GeoMap-Bench[3], where the principles are replaced with those recommended by the USGS [4]. The results show that MEDA can also improve performance in specialized non-medical domains.
>
> ||GeoMap-Bench|
> |-|-|
> |LLaVA-1.5|12.6|
> |+Ours|14.6|
>
>
> Moreover, for general multimodal **domains with less rigid reasoning templates** (e.g. natural image reasoning), **MEDA's global-local-summary reasoning chain in PDI is also applicable**.  Similarly, we evaluate MEDA on 100 sampled questions from GQA [5] while retaining only the global-local-impression structure. The resulting 1.4% improvement over LLaVA-1.5 suggests that MEDA remains beneficial for general multimodal reasoning.
>
> ||GQA|
> |-|-|
> |LLaVA-1.5|60.3|
> |+Ours|61.6|
>
>
> [1]Shai A S, et al. Transformers represent belief state geometry in their residual stream[J]. NeurIPS, 2024.
>
> [2]Wehner J, et al. Taxonomy, opportunities, and challenges of representation engineering for large language models[J]. TMLR
>
> [3]Huang Y, et al. Peace: Empowering geologic map holistic understanding with mllms[C]. CVPR 2025
>
> [4] Earth Resources Observation and Science (EROS) Center. Mapping Land Use and Land Cover[EB/OL]. U.S. Geological Survey, [2026-03-28][https://www.usgs.gov/centers/eros/mapping-land-use-and-land-cover](https://www.usgs.gov/centers/eros/mapping-land-use-and-land-cover)
>
> [5]Hudson D A, Manning C D. Gqa: A new dataset for real-world visual reasoning and compositional question answering[C]. CVPR 2019

---

> > ### Author Rebuttal · Reviewer_YjJS · 2026-04-01
> >
> > Thanks for the response, I will maintain my score.

---

> > > ### Author Response · Authors · 2026-04-01
> > >
> > > Thank you very much for taking the time to review our rebuttal. If you have any further questions, please feel free to contact us at any time.

---

### Official Review · Reviewer_Uozx · 2026-03-11

**Soundness:** 3
**Presentation:** 3
**Significance:** 3
**Originality:** 3
**Overall Recommendation:** 4
**Confidence:** 3

**Summary:**

This paper proposes MEDA, an inference-time activation editing method for mitigating hallucinations in Medical Large Vision-Language Models. The core insight is that existing generic activation editing methods rely solely on visual appearance contrasts and cannot capture the domain-specific knowledge clinicians depend on. MEDA introduces two complementary components to address this. Query-decisive Manifestation Steering retrieves query-relevant imaging manifestations from authoritative medical databases to construct manifestation-aware steering vectors. Principle-driven Diagnosis Induction encodes textbook-recommended diagnostic reading principles into structured templates to guide activation editing. These two vectors are jointly applied during inference. Experiments across six datasets, three imaging modalities, and six LVLMs demonstrate improvements over both generic and specialized medical baselines, with up to 10.6% accuracy gain on IU-Xray and strong few-shot and cross-dataset generalization.

**Compliance With Llm Reviewing Policy:**

Affirmed.

**Key Questions For Authors:**

1.How is the head budget K divided between QMS and PDI in practice? Is the split fixed or determined jointly by l2-norm ranking across both sets? Was the split itself tuned on validation data?

2.Entity extraction and scope labeling require GPT-4 API calls for all vector construction samples. What is the actual cost of this preprocessing at scale? How much does performance drop when the smallest open-source model is used for all LLM calls?

3. Did the authors measure retrieval success rates across modalities? A lower match rate for ophthalmology or pathology would help explain performance differences and clarify where the method is most reliable.

**Limitations:**

The authors acknowledge that MEDA requires access to model activations and therefore cannot be applied to closed-source LVLMs. This is an honest and important disclosure. Two additional limitations deserve mention. First, the retrieval databases may not cover rare conditions or non-Western disease presentations, introducing systematic gaps in certain clinical settings. Second, the GPT-4 dependency affects reproducibility in API-restricted environments. Addressing these points would improve the paper's transparency.

**Strengths And Weaknesses:**

**Strengths**

1.The problem motivation is compelling. Generic activation editing methods derive steering directions from image-level visual contrasts and cannot reliably encode expert semantic knowledge. The distinction between "what a disease looks like" and "how a clinician reasons about an image" is non-trivial and drives the two-component design in a coherent way.

2.The technical execution is mostly solid. The category-aware entity extraction places disease above anatomy above other entities, which is a clinically reasonable heuristic. The use of Bio-ClinicalBERT for entity-to-term alignment and threshold-controlled retrieval are practical design decisions. PDI's scope-conditioned template decomposes reports into global assessment, local evidence, and diagnostic summary, mapping cleanly onto established radiological reading principles.

3.Experimental coverage is thorough. The authors compare against training-based, RAG-based, and inference-time editing baselines across six datasets, three modalities, and six LVLMs of varying sizes. The few-shot analysis down to 10 samples and cross-dataset generalization experiments address realistic deployment scenarios. The reported 34.1x speedup over MMedRAG in vector construction is meaningful for clinical application.

**Weaknesses**

1.The method depends on GPT-4 for entity extraction and scope classification. The authors show some robustness across model choices in Figure 5, but the comparison is limited to IU-Xray with narrow absolute differences. How the system degrades without access to a capable LLM is not analyzed. The paper claims low-cost deployment as a practical advantage, but this claim sits uncomfortably alongside a mandatory GPT-4 preprocessing step.

2.The cooperative steering mechanism is underspecified. The paper allocates K=32 edited heads split between QMS and PDI by l2-norm ranking, but does not explain how the budget is divided between the two modules. Whether the two head sets can overlap and how conflicts would be handled is not discussed.

2.The broad multi-modality framing is somewhat overstated. The primary analysis focuses on chest X-ray radiology. The ophthalmology and pathology results cover only two datasets and are compared only against generic editing baselines without training-based or RAG comparisons.

3.Beyond two case studies, no qualitative error analysis of generated reports is provided. It would be useful to know whether remaining hallucinations follow systematic patterns and whether MEDA's two components address different error types.

---

> ### Author Rebuttal · Authors · 2026-03-31
>
> We greatly appreciate your constructive suggestions and favorable comments on MEDA's motivation, technical execution and experiments.
>
> > ***W1 & Q2:** ... depends on GPT-4 ... the comparison is limited to IU-Xray ... . How the system degrades without access to a capable LLM is not analyzed.  ... claims low-cost deployment ... uncomfortably alongside a mandatory GPT-4 preprocessing* *... the actual cost of this preprocessing ... performance drop when the smallest open-source model is used ...?*
>
> First, as shown in Figure 6, **MEDA remains a low-cost method when GPT-4 preprocessing is included, with a total cost 34.1× lower than** that of training-based methods.
>
> Second, MEDA can **use open-source LVLMs for preprocessing with comparable performance, thus eliminating the mandatory need for GPT-4.** Beyond IU-Xray, replacing GPT-4o(200B) with sufficiently lightweight Qwen2.5-VL-Instruct (8B, as scale-similar to Med-LVLMs) changes ACC by only 0.3% on average across three datasets, with negligible impact on MEDA's overall gains.
>
> ||MIMIC-CXR|Harvard|PMC-OA|
> |-|-|-|-|
> |Baseline|72.5|85.6|60.4|
> |+GPT-4o|79.4|88.3|65.5|
> |+Qwen2.5-VL|79.0|88.1|65.2|
>
> > ***W2 & Q1:** ... does not explain how the budget is divided ... Whether ... can overlap and how conflicts would be handled is not discussed.*
>
> We will further clarify in Section 3.4 that **the head budget K is allocated by joint l2-norm ranking across both sets** without validation tuning. Specifically, all 2048 vectors(1024 from QMS and 1024 from PDI) are jointly ranked, with the top K vectors being selected for steering.
>
> Currently, **our strategy allows head overlap. Conflicts are unlikely to hurt performance, both vector types are expected to steer in a similar truthful direction.** However, we agree with your insightful suggestion and will further study this issue, e.g. devising a conflict detector.
>
> > ***W3:**  ... focuses on chest X-ray radiology. The ophthalmology and pathology results cover only two datasets ... without training-based or RAG comparisons.*
>
> - **Primlary Focus: Most research[1-3] have focused on Chest X-ray radiology driven by abundant public datasets[2].** Therefore, we focus our experiments on this modality to better highlight MEDA' s advantage over strong training-based and RAG baselines
> - **Other modalities and datasets：Following one of the few multi-modality method, MMedRAG[3], we further evaluate ophthalmology and pathology on representative datasets**. The results show that MEDA also outperforms MMedRAG in these modalities.
>
> ||Harvard|PMC-OA|
> |-|-|-|
> |LLaVA-Med-1.5|85.6/30.3|60.4/29.9|
> |+MMedRAG|87.4/32.0|63.7/33.1|
> |+Ours|88.3/32.1|65.5/34.2|
>
> - **Without training-based or RAG comparisons:** Notabley, we cannot compare these methods on the ophthalmology and pathology, as **they do not provide the essential training instructions(Exgra-Med) and preference training data(RULE and MMedPO) required for these specialties.**
>
> > ***W4:** ... no qualitative error analysis of generated reports is provided. ... whether remaining hallucinations follow systematic patterns and whether two components address different error types.*
>
> **The remaining hallucinations suggests that MEDA can be further improved for non-critical findings in Med-LVLMs**. For example ,both the baseline and MEDA omit  "The aorta is mildly tortuous and ectatic." in GT, likely because the Med-LVLM prioritizes the lungs and heart.
>
> **Under the MedHEval[1] taxonomy on MIMIC-CXR, QMS is more effective for knowledge deficiency as QMS improves manifestation elicitation, while PDI better addresses context misalignment** by promoting more contextually grounded reasoning. We will add this analysis to the revised paper
>
> ||knowledge deficiency|context misalignment|
> |-|-|-|
> |LLaVA-Med-1.5|64.6|80.5|
> |+ QMS|72.0|82.8|
> |+ PDI|69.9|86.3|
>
> > ***Q3:** ... measure retrieval success rates across modalities? ... help explain performance differences and clarify where the method is most reliable.*
>
> Following your advice, we evaluate retrieval success rate using GPT-4o. Radiology, ophthalmology, and pathology achieve high success rates of 95.3%, 97.1%, and 96.8%, suggesting **retrieval quality is not directly correlated with performance differences.**
>
> We attribute the smaller gain in ophthalmology (+2.7 vs. >5.1 in others) to two factors: **limited ophthalmology coverage in current Med-LVLM pre-training** that weakens QMS, and **relative simplicity of ophthalmic images (high baseline ACC)**, which reduces the need for PDI.
>
> Overall, MEDA is most reliable **where knowledge encoding is robust and expert reasoning is required**. We will further study specialty-wise conditions to better define its reliability.
>
> [1]Taxonomy, opportunities, and challenges of representation engineering for large language models. TMLR
>
> [2]A survey of deep-learning-based radiology report generation, Medical Image Analysis, 2025
>
> [3]Mmed-rag: Versatile multimodal rag system for medical vision language models. ICLR 2025.

---

> > ### Author Rebuttal · Reviewer_Uozx · 2026-04-08
> >
> > Thanks for the response, I will maintain my score.

---

> > > ### Author Response · Authors · 2026-04-08
> > >
> > > Thank you very much for taking the time to review our rebuttal. If you have any further questions, please feel free to contact us at any time.

---

### Official Review · Reviewer_QQcB · 2026-03-12

**Soundness:** 2
**Presentation:** 3
**Significance:** 2
**Originality:** 3
**Overall Recommendation:** 4
**Confidence:** 4

**Summary:**

Existing activation editing methods show promise in mitigating hallucinations but are not well-suited for clinical tasks. This work proposes MEDical-oriented Activation Editing (MEDA), which integrates Query-decisive Manifestation Steering (QMS) and Principle-driven Diagnosis Induction (PDI) to improve medical expertise utilization in Med-LVLMs. Specifically, QMS retrieves authoritative, query-decisive imaging manifestations to serve as trusted guidance for activation steering, while PDI constructs principle-embedded diagnostic prompts to induce expert-like clinical reasoning. Experiments across diverse modalities demonstrate that MEDA effectively boosts response factuality for both VQA and report generation.

**Compliance With Llm Reviewing Policy:**

Affirmed.

**Final Justification:**

Thanks for the detailed response. My concerns have been addressed, and I will maintain my positive rating.

**Key Questions For Authors:**

1.	This work leverages trusted samples for activation editing. It would be valuable to compare this with a baseline where the same samples are used as few-shot in-context examples. This would help clarify whether the performance gain comes from the editing mechanism itself or just the additional knowledge.
2.	While quantitative ablation results are provided, a qualitative analysis of the model's behavior when using only QMS or PDI would be insightful.
3.	Could the authors discuss whether MEDA can be extended to incorporate other forms of medical expertise, beyond manifestation-level knowledge and diagnostic principles? Are there limitations in extending this approach to other medical specialties?
4.	The strategy of using multiple decomposed editing vectors has been explored in prior LVLM research, e.g., SHARP [1]. It would be better to add a brief discussion comparing with SHARP to better position MEDA and strengthen the related work section.

[1] Wu J, Ding Y, Liu G, et al. Sharp: Steering hallucination in lvlms via representation engineering[C].

**Limitations:**

yes

**Strengths And Weaknesses:**

1.	This paper introduces a novel activation editing framework that effectively integrates professional medical knowledge to steer Med-LVLMs. The proposed methodology, which combines query-decisive manifestation retrieval with principle-driven diagnostic induction, is both theoretically reasonable and technically sound.
2.	The experiments are comprehensive and demonstrate the effectiveness of the proposed method. MEDA not only achieves superior performance but also maintains high computational efficiency.
3.	The paper is well-written and logically structured.

---

> ### Author Rebuttal · Authors · 2026-03-31
>
> Thank you very much for supporting our methodology and giving valuable feedback.
>
> > ***Q1:** This work leverages trusted samples for activation editing. It would be valuable to compare this with a baseline where the same samples are used as few-shot in-context examples. This would help clarify whether the performance gain comes from the editing mechanism itself or just the additional knowledge.*
>
> Thank you for your insightful comments. Prior studies[1,2] have revealed that the **editing mechanism works by constructing external positive and negative sample pairs to elicit internal knowledge encoded in truthful activations.** Accordingly, unlike few-shot in-context learning, MEDA leverages additional knowledge to construct higher-quality positive samples, thereby more precisely probing truthful activations and eliciting internal medical knowledge.
>
> Following your suggestion, we compare MEDA against a simply few-shot in-context learning baseline by sampling from the same construction data of MEDA. Results show that this few-shot baseline yields only marginal improvement, indicating that MEDA's advantage indeed stems from the editing mechanism.
>
> |Methods|IU-Xray|Harvard-FairVLMed|PMC-OA|
> |-|-|-|-|
> |LLaVA-Med-1.5|74.2/50.8|60.4/29.9|85.6/30.3|
> |+ positive text|75.6/52.1|60.9/28.8|86.0/30.6|
> |+ Ours|**84.8/79.7**|**65.5/34.2**|**88.3/32.1**|
>
> > ***Q2:** While quantitative ablation results are provided, a qualitative analysis of the model's behavior when using only QMS or PDI would be insightful.*
>
> Thanks for your meaningful suggestions. Specifically, we find that **QMS is more effective at identifying critical findings.** For example, QMS enables LLaVA-Med-1.5 to generate findings "opacities present in the right costophrenic angle", suggesting that it helps elicit the internal medical knowledge needed to recognize subtle opacities that might otherwise be overlooked.
>
> In addition, **reports produced with PDI more often reflect a holistic interpretation of the entire image.** For instance, PDI frequently corrects mistaken statements "pleural effusion is observed" to correct interpretations "The lungs appear to be clear." We attribute this behavior to PDI’s ability to elicit comprehensive diagnostic reasoning, allowing the model to suppress distracting regional cues and arrive at a more faithful overall interpretation. We will include this qualitative analysis of QMS and PDI in the revised paper.
>
> > ***Q3:** Could the authors discuss whether MEDA can be extended to incorporate other forms of medical expertise, beyond manifestation-level knowledge and diagnostic principles? Are there limitations in extending this approach to other medical specialties?*
>
> Thank you for your constructive comments. Our MEDA **is a flexible editing framework that can use any truthful clinical information as positive guidance**, making it easy to incorporate other forms of medical expertise and extend to other medical specialties. For specialties with expert knowledge or diagnostic principles, MEDA can be naturally extended by replacing the corresponding sources in QMS and PDI with specialty-specific ones. Moreover, MEDA's general strategy(e.g. global-local-summary manner in PDI) is also applicable across most specialties. Apart from radiology in our main experiments, we also evaluate pathology and ophthalmology specialties, and the results in Table 2 further demonstrate its scalability.
>
> One **potential limitation** in extending MEDA to other medical specialties lies in **constructing high-quality and operationalized expert data sources**. We plan to further explore scalable and generalizable ways to construct such expert data sources in future work.
>
> > ***Q4:** The strategy of using multiple decomposed editing vectors has been explored in prior LVLM research, e.g., SHARP. It would be better to add a brief discussion comparing with SHARP to better position MEDA and strengthen the related work section.*
>
> Thank you for your valuable feedback. Since SHARP doesn't provide open-source code or datasets, we did not compare it in the experiments due to concerns about reproducibility. **We will strengthen our related work section in the revised paper by providing the comparison discussion between MEDA and SHARP.**
>
> By targeting **textual priors** (biased co-occurrence patterns learned during pretraining) and **visual-text conflicts** (inconsistencies between the query and the image), **SHARP mitigates hallucinations in general LVLMs** using two decomposed editing vectors. In contrast, **MEDA is better suited to medical hallucinations** by precisely eliciting the internal medical and diagnostic knowledge of Med-LVLMs. Therefore, SHARP and MEDA are designed to address different hallucination mechanisms in different domains.
>
> [1]Li K, et al. Inference-time intervention: Eliciting truthful answers from a language model[J]. NeurIPS 2023.
>
> [2]Rimsky N, et al. Steering llama 2 via contrastive activation addition[C]. ACL 2024

---

> > ### Author Rebuttal · Reviewer_QQcB · 2026-04-01
> >
> > Thanks for the detailed response. My concerns have been addressed, and I will maintain my positive rating.

---

> > > ### Author Response · Authors · 2026-04-01
> > >
> > > Thank you very much for taking the time to review our rebuttal. If you have any further questions, please feel free to contact us at any time.

---

### Official Review · Reviewer_Krdz · 2026-03-12

**Soundness:** 3
**Presentation:** 3
**Significance:** 2
**Originality:** 3
**Overall Recommendation:** 4
**Confidence:** 3

**Summary:**

This paper adapts activation editing to the medical imaging setting for hallucination mitigation. Instead of relying only on coarse visual contrasts, it injects two forms of medical knowledge—query-decisive manifestations and principle-driven diagnostic guidance—into steering vector construction, with the goal of eliciting more medically grounded responses from Med-LVLMs.

**Compliance With Llm Reviewing Policy:**

Affirmed.

**Final Justification:**

The rebuttal addressed my concerns, so I raised my score.

**Key Questions For Authors:**

1. Does MEDA still provide clear gains on stronger and more recent Med-LVLMs? A positive answer would substantially strengthen the paper’s broader relevance.

2. Can the response provide stronger controls to separate the effect of medical-oriented activation editing from the effect of simply adding more helpful positive text? This would clarify the mechanism behind the improvement.

3. What types of hallucinations are reduced most by MEDA? A finer-grained error analysis would make the contribution more interpretable.

4. How do the response verify that PDI improves diagnostic reasoning rather than mainly encouraging answer-style imitation?

**Limitations:**

No. The paper would benefit from a more explicit discussion of its limited coverage of stronger contemporary Med-LVLMs, the possibility that some gains come from external guidance rather than true knowledge elicitation, and the robustness risks introduced by its multi-stage pipeline.

**Strengths And Weaknesses:**

Strengths:
1. The paper studies an important problem and proposes a clear medical-oriented extension of activation editing.

2. Using query-guided entity extraction to retrieve more relevant medical knowledge is a sensible design choice.

3. The principle-driven component provides an interesting way to inject structured clinical priors during inference.


Weaknesses:
1. The empirical validation relies heavily on LLaVA-Med-1.5 as the main experimental base model, making it unclear whether the method would still provide comparable gains on substantially stronger Med-LVLMs. More broadly, the Med-LVLMs included in the Baseline and Comparative experiments do not sufficiently represent the latest generation of Med-LVLMs, which limits the representativeness of the evaluation.

2. The paper does not sufficiently disentangle whether the gains come from better elicitation of internal medical knowledge or from providing additional external textual guidance to weaker base models.

3. The method depends on multiple upstream components, including entity extraction, retrieval, report decomposition, and template design, which may introduce compounded errors.

4. It is not yet clear whether the PDI component improves reasoning itself or mainly steers the model toward answer-style imitation.

---

> ### Author Rebuttal · Authors · 2026-03-31
>
> We sincerely thank you for your detailed review and positive feedback on our methodology.
>
> > ***W1 & Q1:** ... relies heavily on LLaVA-Med-1.5 ... whether the method would still provide comparable gains on stronger and more recent Med-LVLMs ? ... do not sufficiently represent the latest generation of Med-LVLMs, ...*
>
> Beyond LLaVA-Med-1.5, Figure 3 also shows that **MEDA yields at most 15.3% gain in ACC and 14.6% gain in RaTEScore across five more recent Med-LVLMs, including the latest LLaVA-Tri (ICLR 25)[1], Qwen2.5-VL-Instruct (released in January 2025)[2]**, etc. We further compare against AOR(NeurIPS 25)[3], a more recent Med-LVLM **fully released in January 2026**. The results (ACC/RaTEScore) show that META achieves clear gains of at most 12.2%, outperforming other methods.
>
> |Methods|IU-Xray|Harvard|PMC-OA|
> |-|-|-|-|
> |AOR|57.0/53.1|21.3/18.3|41.9/31.2|
> |+ MMedRAG|59.3/60.2|22.9/22.9|44.5/36.9|
> |+ AFTER|56.8/58.3|22.7/22.3|45.2/35.9|
> |+ Ours|**63.3/65.3**|**24.5/25.2**|**45.5/37.3**|
>
>
> > ***W2 & Q2:** ... does not sufficiently disentangle whether the gains come from better elicitation of internal medical knowledge or from providing additional external textual guidance ... provide stronger controls to separate the effect of ...*
>
> Prior studies[4,5] have shown that activation editing **works by constructing external positive and negative sample pairs, thereby contrasting paired activations and steering the LVLM to elicit internal knowledge encoded in truthful activations.** Similarly, MEDA uses external textual guidance to construct higher-quality positive samples, thereby more precisely probing truthful activations and eliciting internal medical knowledge, without injecting extra helpful text.
>
> We further compare MEDA on AOR[3] with adding MEDA's construction samples as positive text during inference. This baseline yields only marginal gains, supporting our claim.
>
> |Methods|IU-Xray|Harvard|PMC-OA|
> |-|-|-|-|
> |AOR|57.0/52.1|21.3/18.3|41.9/31.2|
> |+ positive text|57.2/51.6|20.8/17.9|42.5/32.0|
> |+ Ours|**63.3/65.3**|**24.5/25.2**|**45.5/37.3**|
>
>
> > ***W3:**  The method depends on multiple upstream components, including ..., which may introduce compounded errors.*
>
> First, **MEDA devises multiple high-precision selection mechanisms across most upstream components** (e.g., priority function , relevance filtering in Section 3.2) **to keep compounded errors within a controllable range.**
>
> Moreover, prior works[4,5] have shown that **statistical averaging in steering vector construction effectively mitigates the impact of errors from individual samples**.
>
> > ***W4 & Q4:**  ... whether the PDI component improves reasoning itself or mainly steers the model toward answer-style imitation. How do the response verify ... ?*
>
> **PDI is not intended to induce template-style imitation, but to construct higher-quality positive samples that elicit internal reasoning activations**. As discussed in response to W2, activation editing cosntructs external sample pairs for contrastive steering to elicit internal knowledge. Accordingly, the PDI template serves as an informative positive sample, rather than a stylistic target.
>
> We further ask GPT-4o to rate the stylistic similarity between steered 50 VQA, 50 report answers and the PDI's templates on a 10-point scale. The only 1.05 and 4.21 average scores far below the threshold, supporting our claim.
>
> > ***Q3:** What types of hallucinations are reduced most by MEDA? A finer-grained error analysis would make ....*
>
> As shown in Appendix B.1,**we follow the hallucination taxonomy of the MedHEval**[6] for radiology evaluation, and we will add finer-grained error analysis in revised paper.
>
> We report ACC on IU-Xray for visual misinterpretation and on MIMIC-CXR for the other categories, excluding MIMIC-CXR visual misinterpretation to avoid data leakage (Appendix B.1). MEDA performs best on Symptom and knowledge deficiency from QMS-provided knowledge, and improves context misalignment by PDI-induced context-aware reasoning, but shows smaller gains on Measurement and Radiology Features, likely because these rely less on the capabilities elicited by MEDA.
>
> |Methods|visual misinterpretation||||knowledge deficiency|context misalignment|
> |-|-|-|-|-|-|-|
> ||Anatomy|Measurement|Symptom|Radiology Features|||
> |LLaVA-Med-1.5|88.9|70.3|69.5|93.6|64.6|80.5|
> |+ Ours|92.3(+3.4)|71.5(+1.2)|78.9(+9.4)|94.8(+1.2)|72.3(+7.7)|86.5(+6.0)|
>
> [1]MedTrinity-25M: A Large-scale Multimodal Dataset with Multigranular Annotations for Medicine. ICLR 2025.
>
> [2]Qwen2.5-vl Technical report. arXiv:2502.13923.
>
> [3]AOR: Anatomical Ontology-Guided Reasoning for Medical Large Multimodal Model in Chest X-Ray Interpretation. NeurIPS 2025.
>
> [4]Inference-time intervention: Eliciting truthful answers from a language model. NeurIPS 2023.
>
> [5]Steering llama 2 via contrastive activation addition. ACL 2024
>
> [6]Medheval: Benchmarking hallucinations and mitigation strategies in medical large vision-language models. arXiv:2503.02157.

---

> > ### Author Rebuttal · Reviewer_Krdz · 2026-04-01
> >
> > Thank you for the detailed rebuttal. This rebuttal addresses most of my main concerns in a direct way and also provides useful additional empirical evidence, so I will raise my score moderately.
> >
> > However, my main reservation still remains: the current evidence is still insufficient to fully establish MEDA’s applicability boundary on stronger, newer-generation Med-LVLMs. Although the authors added results on AOR, its underlying backbone is still LLaVA-1.5. Therefore, the current results more strongly support that the method is effective on a set of existing Med-LVLMs, rather than demonstrating that its gains would continue to hold stably on substantially stronger and more recent models. I understand that some stronger recent medical models in the community, such as Lingshu and Hulu-Med, are still mainly available as preprints or technical reports, so it is understandable that they were not included in the experiments. Still, if future validation on such newer models becomes possible, it would be very helpful for more clearly defining the applicability boundary and longer-term value of MEDA.
> >
> > In addition, I noticed that the newly added AOR results in the rebuttal appear somewhat different from its relative performance reported in the original paper: AOR appears stronger than LLaVA-Med on several chest X-ray benchmarks in its own paper, whereas it appears weaker on IU-Xray here. Since these results may correspond to different datasets, task formulations, and evaluation protocols, it would be helpful if the authors could further clarify whether the rebuttal results for AOR were obtained under its original evaluation setting or re-evaluated under the paper’s unified protocol. Such clarification would help interpret this difference more accurately. At the same time, since AOR appears weaker than LLaVA-Med under the rebuttal’s IU-Xray setting, the added AOR results only partially alleviate my concern: they show that MEDA is also effective on another Med-LVLM, but do not yet strongly establish its applicability to stronger, newer-generation models.
> >
> > **[Updated on 7th April]**
> >
> > **Thanks for the second round of rebuttal. These new results have addressed my remaining concerns.**

---

> > > ### Author Response · Authors · 2026-04-01
> > >
> > > Thank you very much for your valuable feedback and for raising your score! Similar to other medical hallucination mitigation research, we primarily focus on Med-LVLMs that are published in top conferences and journals. We fully appreciate your suggestion, and **we have evaluated the ACC and RaTEScore of MEDA on the latest Lingshu-I-8B model**:
> > >
> > > |Methods|IU-Xray|PMC-OA|
> > > |-|-|-|
> > > |Lingshu-I-8B|79.3/58.6|63.2/38.6|
> > > |+ MMedRAG|82.1/60.6|65.1/42.2|
> > > |+ AFTER|81.6/58.1|65.9/38.9|
> > > |+ Ours|**84.8/64.5**|**68.9/45.6**|
> > >
> > > These results show that MEDA remains effective on the newer and stronger Lingshu-I-8B (demonstrates better empirical performance than LLaVA-v1.5), outperforming SOTA method AFTER and MMedRAG on both IU-Xray and PMC-OA. In particular, compared with the base Lingshu-I-8B model, MEDA improves the two metrics by 5.5%/5.9% on IU-Xray and 5.7%/7.0% on PMC-OA, further demonstrating its effectiveness and generalizability on more advanced Med-LVLMs.
> > >
> > > Regarding the issue of the AOR baseline’s relatively weaker performance on IU-Xray compared to the results reported in the original paper, we believe this may be due to the following reasons:
> > > * As outlined in Section 4.1, the questions used for testing on IU-Xray were provided by MedHEval. To test more complex, multi-type hallucination issues, these questions may be more challenging than those in the original IU-Xray dataset.
> > > * Since the official GitHub code of AOR did not provide the inference format or templates, and the weights were not available on Hugging Face along with the sample inference code, we used the same template as LLaVA-1.5 for our experiments. The performance may vary to some extent due to differences from the optimal template. Unfortunately, due to time constraints, we were unable to retrieve the prompt that would yield optimal results. However, we ensured consistency in the prompts across MEDA, baseline, and AFTER in our comparative experiments to ensure a fair comparison and to highlight the effectiveness of MEDA.
> > >
> > > We will incorporate the added experiments and analysis into the revised version to further improve our paper.
> > >
> > > **[Updated on 6th April]**
> > > Dear Reviewer Krdz, we hope this message finds you well. On April 3, we added the supplementary comparative experiments on Lingshu-I-8B, **which further validate the effectiveness of MEDA on a stronger Med-LVLM**. As the rebuttal deadline is approaching, we would be sincerely grateful if you could let us know whether these new results, together with our response regarding the IU-Xray experiments, have adequately addressed your remaining concerns. Should you have any further questions, please feel free to let us know. We greatly appreciate your valuable feedback and thank you again for your time and consideration.

---

### Decision · Program_Chairs · 2026-04-30

**Decision:**

Accept (regular)

**Comment:**

This paper proposes MEDA, a medical-oriented activation editing method for hallucination mitigation in Md-LVLMs. All four reviewers gave unanimous scores of weak accept, and all marked their concerns as fully resolved after rebuttal. Reviewers recognized the compelling problem motivation, solid technical execution, thorough experimental coverage across six benchmarks and six LVLMs, and impressive efficiency. During rebuttal, the authors added experiments on stronger Med-LVLMs, demonstrated that gains come from the editing mechanism rather than external text injection, provided finer-grained hallucination type analysis, and showed generalizability to non-medical domains. Given the unanimous positive consensus and thorough rebuttal addressing all reviewer concerns, I recommend accept.